# Glycemic Control in Critically Ill COVID-19 Patients: Systematic Review and Meta-Analysis

**DOI:** 10.3390/jcm12072555

**Published:** 2023-03-28

**Authors:** Subhash Chander, Vishal Deepak, Roopa Kumari, Lorenzo Leys, Hong Yu Wang, Puja Mehta, FNU Sadarat

**Affiliations:** 1Department of Internal Medicine, Mount Sinai Beth Israel, New York, NY 10003, USA; 2Department of Internal Medicine, Section of Pulmonary, Critical Care, and Sleep Medicine, School of Medicine, West Virginia University, Morgantown, WV 26506, USA; 3Department of Pathology, Mount Sinai Morningside, and Mount Sinai West, New York, NY 10025, USA; 4Department of Medicine, Section of Pulmonary and Critical Care Medicine, Mount Sinai Beth Israel, New York, NY 10003, USA; 5Department of Medicine, Section of Pulmonary and Critical Care Medicine, Mount Sinai Morningside, and Mount Sinai West, New York, NY 10025, USA; 6Department of Internal Medicine, Section of Nephrology, Yale School of Medicine, New Haven, CT 06510, USA; 7Department of Internal Medicine, University of Buffalo, New York, NY 14215, USA

**Keywords:** glycemic control, COVID-19, critically ill patients, hyperglycemia, tight blood sugar control

## Abstract

**Background:** Given the mortality risk in COVID-19 patients, it is necessary to estimate the impact of glycemic control on mortality rates among inpatients by designing and implementing evidence-based blood glucose (BG) control methods. There is evidence to suggest that COVID-19 patients with hyperglycemia are at risk of mortality, and glycemic control may improve outcomes. However, the optimal target range of blood glucose levels in critically ill COVID-19 patients remains unclear, and further research is needed to establish the most effective glycemic control strategies in this population. **Methods:** The investigation was conducted according to the Preferred Reporting Items for Systematic Reviews and Meta-Analyses (PRISMA). Data sources were drawn from Google Scholar, ResearchGate, PubMed (MEDLINE), Cochrane Library, and Embase databases. Randomized controlled trials, non-randomized controlled trials, retrospective cohort studies, and observational studies with comparison groups specific to tight glycemic control in COVID-19 patients with and without diabetes. **Results:** Eleven observational studies (26,953 patients hospitalized for COVID-19) were included. The incidence of death was significantly higher among COVID-19 patients diagnosed with diabetes than those without diabetes (OR = 2.70 [2.11, 3.45] at a 95% confidence interval). Incidences of death (OR of 3.76 (3.00, 4.72) at a 95% confidence interval) and complications (OR of 0.88 [0.76, 1.02] at a 95% confidence interval) were also significantly higher for COVID-19 patients with poor glycemic control. **Conclusion:** These findings suggest that poor glycemic control in critically ill patients leads to an increased mortality rate, infection rate, mechanical ventilation, and prolonged hospitalization.

## 1. Introduction

In 2020, the world was hit by the unprecedented SARS-CoV2 pandemic, which pushed scientists to quickly understand the virus, the human immune system, and its response to COVID-19 infection. Early on during the study of the virus and its infection, multiple epidemiological studies proved that elevated blood sugar is a significant risk factor for mortality in COVID-19 patients [1,2]. Before the pandemic in the 2000s, diabetes was already a known risk factor for negative results in other disease epidemics, such as SARS-CoV-1 infections, in which hyperglycemia was regarded as an independent marker of mortality [3]. The prevalence of hyperglycemia among COVID-19 patients does not significantly differ from that of the general population, suggesting that the conditions do not increase the risk of contracting COVID-19 infection but can substantially make the outcomes worse [2,4,5,6]. Multiple clinical studies have shown that high blood sugar is a major risk factor for increased fatality and severity of COVID-19 infection [7,8]. COVID-19 patients with uncontrolled hyperglycemia are more likely to end up in the ICU, and the mortality rate is estimated to be approximately three times higher than that of patients without hyperglycemia [9]. According to previously published literature, hyperglycemic patients, whether they have diabetes or not, have an increased risk of mortality, morbidity, and hospitalization; however, better blood glucose control can help improve clinical outcomes [10,11,12].

Multiple mechanisms (glycemic and hyperglycemic control, inflammatory response, and impaired immune function due to hyperglycemia) have been linked to worse COVID-19 prognosis over time [13,14]. Poor or a lack of glycemic control has been linked to worse outcomes, such as prolonged hospitalization, higher health costs, higher mortality and morbidity, and multi-organ injuries [6,15]. Hyperglycemic patients’ conditions (with or without diabetes) harm the efficacy of coronavirus therapies such as tocilizumab [8,16]. Although many other mechanisms lack scientific explanation, optimization of glycemic control can significantly improve COVID-19 outcomes. Well-controlled blood glucose levels (70–180 mg/dL) reduce clinical intervention, all-cause mortality, and major organ deterioration [17]. According to Fadini et al., for every 10 mg/dL decrease in sugar levels between hospitalization and 18 days, there is a relative drop of 11% in severe infection risk in glycemic patients [4].

The limited literature suggests that blood glucose control in COVID-19 inpatients is inadequate. This could be due to inflammation, high-stress levels, insulin resistance, and a cytokine-mediated state with poor blood glucose control measures [14,18,19]. Scientists also note the likelihood of SARS-CoV-2 penetrating pancreatic islets, which can cause damage to beta cells and hence insulin deficiency [5,20]. Such conditions can significantly worsen glycemic control or trigger acute hyperglycemia in patients with normal blood sugar levels. In contrast, reports show a significant number of hypoglycemic cases from inpatients, most likely due to anorexia, one of the COVID-19 symptoms, without first adapting to drugs used to lower glucose levels [13,21]. Other factors are also linked to blood glucose fluctuation, particularly in COVID-19 patients, including the use of glucocorticoids that can cause glycemic excursions; thus, their impact should be considered during insulin pattern setup [22]. Additionally, health practitioners may encounter situations that require careful attention. Specialists from all departments might find themselves dealing with COVID-19 patients outside their usual line of work; thus, all clinicians must learn how to manage hyperglycemia [23,24,25].

Identifying the risk factors for SARS-CoV-2 infection progression and fatality provides substantial evidence supporting appropriate clinical control and the optimized allocation of medical resources. Hyperglycemia is a crucial risk factor for COVID-19 progression; hence, extensive investigations are needed. The above literature review implies that high blood glucose levels are a major determinant of mortality among COVID-19 patients. Given the mortality risk in COVID-19 patients, it is essential to estimate the impact of glycemic control and mortality rates among inpatients to design and implement evidence-based BG control methods for these inpatients. Due to a lack of sufficient data investigating this subject matter, reviewers decided to systematically assess the available published literature and draw conclusions that will help guide future clinical procedures in glycemic management, particularly among critically ill COVID-19 patients. This systematic review and meta-analysis will help us better understand BG management and its influence on COVID-19 inpatients. Therefore, this review will seek to answer the question: “How does glycemic control in critically ill COVID-19 patients affect the treatment outcomes?”. The review was based on previously conducted studies; hence, it may include studies with or without human or animal participants performed in any of the selected clinical studies.

## 2. Materials and Methods

### 2.1. Study Design

This systematic review follows the Cochrane guidelines [26] and is conducted using the Preferred Reporting Items for Systematic Reviews and Meta-Analyses (PRISMA) guidelines [27]. This systematic review and meta-analysis were conducted in accordance with the Preferred Reported Items for Systematic Review and Meta-Analysis (PRISMA) guidelines. The protocol for this systematic review and meta-analysis was registered with the International Prospective Register of Systematic Reviews (PROSPERO) prior to conducting the literature search (registration number CRD42023394275). All steps taken in the review process are reported in accordance with the PRISMA statement.

The approach guides the study search, development of the inclusion criteria, and final study selection processes of the review, subsequently reducing vulnerability (to internal or external sources). The quality of the overall evidence was evaluated using RevMan.

### 2.2. Literature Search

A comprehensive search was performed using the Cochrane Library, PubMed, Embase, ScienceDirect, and Google Scholar electronic databases. Owing to the scarcity of published studies in this area, several databases were selected with additional studies identified from the reference lists of other studies. An experienced reference librarian was included in the design and conduct of the search strategy with input from the principal reviewers. First, a more general search was performed to establish the link between high blood glucose recovery and COVID-19 infection, which produced a variety of studies that did not meet the requirements of the study objectives to answer the research question accurately. Then, within the above-mentioned electronic databases, reviewers used keywords including “glycemic, hyperglycemia, COVID-19, SARS-CoV-2, hyperglycemia management/control, coronavirus, diabetes, blood glucose, and mortality”. Boolean operators (AND/OR) combined the keywords to produce more relevant outcomes within the selected databases. In PubMed, MeSH terms and Boolean operators were used to refine the search terms to generate relevant articles. Additionally, a reputable expert in the field was contacted to confirm the identification of all potentially relevant articles. Additional studies were performed by manually searching reference lists.

### 2.3. Eligibility Criteria

Two independent investigators reviewed all the relevant articles and selected those that were suitable for inclusion. Any disagreement was resolved by reaching a consensus or by a third reviewer. The systematic review applied the PICO to the inclusion and exclusion criteria. All included articles met the following inclusion criteria: (1) population (subjects were COVID-19 patients and hyperglycemic), (2) intervention (management of BG), (3) control (control group whose BG was not controlled), and (4) study outcomes (mortality and complications of coronavirus patients with hyperglycemia). Studies were excluded if they did not have relevant data necessary for the meta-analysis or other issues, such as duplicate publications, editorials, reviews of original studies, position papers, articles without full-text access, letters to the editors, and any study not written in English.

### 2.4. Data Extraction

Working independently, the two reviewers extracted data from different sources. Before extracting the data, the selected studies were assessed for risk of bias according to Cochrane methodological standards [26]. The data extraction phase was conducted by the same researchers involved in the selection of articles. The reviewers extracted specific relevant data using a standardized Excel sheet. The charted variables included authors, location of the study, publication date, study design and population, clinical outcomes, mortality, and COVID-19 complications.

### 2.5. Statistical Analysis

Two types of analyses were used in this investigation. Qualitative (systematic review) and quantitative (meta-analysis) assessments were also performed. A literal analysis was adopted to conduct a systematic review of the evidence provided by all included studies. We then employed Review Manager version 5.4 (RevMan 5.4; The Nordic Cochrane Center, The Cochrane Collaboration, 2014) to perform a meta-analysis of studies reporting quantitative data in line with the outcomes of this review. A fixed-effects model was used to compute effect size. Dichotomous effect sizes were expressed as event rates (ERs) and used to populate the software. The fixed-effects odds ratio (OR) was calculated at a 95% confidence interval (CI), and the level of significance was reported as the *p*-value. Statistical significance was set at *p* < 0.05. Heterogeneity was expressed using the I^2^ statistic and was judged according to the value, which ranged from 0% (complete consistency) to 100% (complete inconsistency). The results of the meta-analysis were reported using a forest plot and publication bias among the analyzed studies was expressed using a funnel plot.

## 3. Results

### 3.1. Study Selection

A total of 229 sources were selected for review; however, 53 articles were excluded based on duplication, and 176 submitted articles were collected for the title and abstract screening. At this stage, the reviewers selected 52 reviews, 23 commentaries, 21 supplementary studies, and 9 irrelevant studies, all of which were excluded from the systematic review. Next, full-text screening was performed for 71 studies, and 57 articles were eliminated because they either did not have a full-text publication or had other eligibility reasons such as studies written in languages other than English. Eleven articles (Table 1.) were selected for inclusion, with an additional three articles added from the reference lists (bibliographies and citations) of previous meta-analyses and systematic reviews. Figure 1 shows the PRISMA 2020 flow [27] with a summary of the selection criteria explained above.

### 3.2. Meta-Analysis Results

Comparison 1: Diabetic vs. Non-Diabetic COVID-19 Patients.

Death

Three [4,25,33] of the eleven included studies reported incidences of death among COVID-19 patients diagnosed with diabetes compared to those without a diabetes diagnosis. We employed a fixed-effects model to calculate the odd ratio of mortality between the two groups and found an OR of 2.70 [2.11, 3.45] at a 95% confidence interval. The current analysis showed a moderately high level of heterogeneity, as evidenced by the I-statistics (I^2^ = 64%). A test for overall effects showed Z = 7.92 (*p* < 0.00001), which is an indicator of a significant difference between the included studies. Figure 2 and Figure 3 present the Forest and funnel plots used for this analysis.

Comparison 2: Well-controlled vs. Poorly Controlled Blood Sugar.

Death

Another set of three studies [3,29,30] compared the populations of COVID-19 patients who had to undergo glycemic control. In this comparison, the analysis sought to compare two groups undergoing different forms of glycemic control, judged as either well-controlled or poorly controlled. Four data comparisons were obtained, and a fixed effects model OR of 3.76 (3.00, 4.72) at a 95% confidence interval was determined. There was a high level of heterogeneity between studies (I^2^ = 93%). The test for overall effect was found to be Z = 11.42 (*p* < 0.00001), which indicates a significant difference between the groups. Figure 4 and Figure 5 show the Forest and funnel plots of this analysis.

### 3.3. Complications

Another set of three studies [29,30,31] that compared the populations of COVID-19 patients who had to undergo glycemic control was assessed. The outcome of this comparison was the incidence of disease complications between the two glycemic control modalities. Four data comparisons were obtained, and a fixed effects model OR of 0.88 [0.76, 1.02] 95% confidence interval was determined. There was a low level of heterogeneity between studies (I^2^ = 20%). The test for the overall effect was found to be Z = 1.70 (*p* < 0.09), which indicated a lack of significant difference between the groups. Figure 6 and Figure 7 present the Forest and funnel plots, respectively, for this analysis.

## 4. Discussion

The results of this meta-analysis indicate that diabetes is a key factor among the most common comorbidities, as it is significantly associated with the severity and mortality of coronavirus infection. The main goal of this study was to determine whether there is a link between glycemic control and death rate or complications in critically ill COVID-19 patients. Healthcare workers use several blood-glucose-lowering medications to treat COVID-19. According to the results obtained from our meta-analysis, glycemic management is heavily linked to mortality and the incidence of severe complications, including ICU admission and COVID-19 in patients with diabetes. This review provides evidence of the impact of glycemic management in this group of patients. However, due to the limitations of studies concerning glycemic control interventions among COVID-19 patients in our meta-analysis, the relationship between various treatments and outcomes in critically ill patients with hyperglycemia and COVID-19 infection needs to be analyzed in more extensive clinical studies.

### 4.1. Glucose Control in Hospital

Insulin therapy remains the most commonly used glucose-lowering medication in critically ill patients. The main aims of treatment are to minimize BG variability, reduce the risk of hypoglycemia, and develop optimal glycemic management. However, there is no ideal treatment protocol for hyperglycemia management in critically ill patients or clear evidence to demonstrate the benefit of one approach over the other(s). A study by Yang et al. indicated that insulin treatment may increase fatality and several complications in patients with diabetes and COVID-19 admitted to the ICU [9]. Although there was no substantial association between insulin treatment and in-hospital admission, the use of insulin to manage blood sugar levels may result in prolonged hospitalization. Continuous intravenous insulin infusion, according to computerized protocols or validated written protocols, is the most effective approach for achieving BG targets [17,32]. However, most healthcare workers prefer protocols that require fewer or no calculations of insulin dosage to those that require such calculations. Training healthcare practitioners on how to efficiently use the protocol and predefine adjustments before changing blood sugar or in any situation in which parenteral nutrition, vasopressors, and corticosteroids, among others, are either added or removed is important for protocol safety [25]. Perez-Belmonte et al. (2020) reported that the use of at-home metformin, insulin, DPP-4i, metformin plus insulin, and metformin plus SGLT-2 was significantly associated with in-hospital mortality, complications requiring ICU admission, mechanical ventilation, and longer hospitalization in COVID-19 patients with diabetes [30]. In contrast, a retrospective investigation by Raoufi et al. indicated that COVID-19 patients with poorly or well-controlled diabetes did not present significant differences in chest CT severity scores and clinical outcomes [29].

### 4.2. Association between Glycemic Control and COVID-19 Outcomes

Patients with COVID-19 and hyperglycemia present with a variety of symptoms, the majority of which can progress to serious complications [9,28,34]. Zhang et al. reported improved clinical outcomes in patients with COVID-19 but with well-controlled BG [9]. The study by Kapoor et al. illustrated the risk of a higher fatality rate with uncontrolled blood sugar levels based on a sample of 5693 hospitalized patients [20]. HbA1C > 7.5% is heavily linked with an increased number of deaths among these patients. The in-hospital fatality rate among patients with more than 70–150 mg/dL (>85 mg/dL) in patients without coronavirus infections was better than that in those with less than 85% [33]. Wang et al. emphasized that poorly managed BG leads to more deaths among patients with diabetes and SARS-CoV-2 infection [34]. The study population involved in specific investigations helps explain these observations. The study included all hospitalized patients, whereas most other studies were limited to ICU patients. Furthermore, the principal mechanisms explaining the link between insulin treatment and poor outcomes among COVID-19 patients with diabetes are unclear; however, there are different explanations for this relationship. Klonoff et al. reported that high blood sugar levels lead to a lower outcome of chronic inflammation, distinguished by adaptive and innate inflammation systems [3]. This condition disrupts the regulation of immune system responses with high levels of pro-inflammatory factors and low levels of anti-inflammatory cytokines. The authors indicated that insulin treatment increased the levels of inflammatory cytokines produced by macrophages in patients with lipopolysaccharide-induced sepsis [35]. Therefore, insulin infusion can rapidly correct complications arising from hyperglycemia, including adhesion molecules and lipid peroxidation. This crucial observation has therapeutic implications for the development of chronic or acute complications in COVID-19 patients with high BG levels.

### 4.3. Diabetes-Associated Complications

Several hypotheses have been proposed to explain the relationship between hyperglycemia and the progression of viral respiratory infections. Hyperglycemia may adversely affect the function of the pulmonary system, suppress immune responses, and increase the production of inflammatory cytokines [17,28]. Furthermore, angiotensin-converting enzyme 2 expression occurs in the pancreas, and it has been suggested that SARS-CoV-2 can cause direct damage to pancreatic islets [17]. However, the influence of BG on the progression of coronavirus infection requires further study. A recent study by Zhu et al. investigated the link between BG control and clinical outcomes in 952 COVID-19 patients with diabetes [33]. The authors found a slightly lower prevalence of diabetes (8.6%) in this population, despite reporting a 13% prevalence among all participants. In another study, the prevalence of diabetes among COVID-19 patients was 8.9%, which presents a huge comparison between the reviewed studies [31]. Similar to the findings of Zhu et al. (17%) [33], approximately 18% of patients with diabetes developed complications, including ARDS [29]. However, there was a major difference between these two studies [29,33] regarding the definitions of poorly controlled and well-controlled BG. Most articles show that COVID-19 patients with well-managed BG have a remarkably lower in-hospital fatality rate than those with poor or uncontrolled BG. The most recent studies on hospitalized patients with COVID-19 and diabetes support the findings of the present meta-analysis. For example, unlike many studies, Mehta et al.’s clinical study did not associate HbA1c values with a higher death rate or poor clinical outcomes such as mechanical ventilation in COVID-19 patients (which contradicts our findings) [36]. Multiple studies have presented that glycemic management may significantly impact the patient’s prognosis with COVID-19 and a co-existing hyperglycemia condition [17,28,29]. An investigation by Wang et al. states that the death rate of in-hospital COVID-19 patients with high BG is approximately 22%, while another study by Zhang et al. was nearly 29% [9,31]. The death rates were significantly higher than those reported in patients with COVID-19 but without diabetes. Bhatti et al. also affirmed the assumption that glycemic management when a patient is admitted to the ICU is directly related to poor clinical outcomes in COVID-19 patients with hyperglycemia [32]. However, ICU admission or mechanical ventilation leading to death was minimal, with reasonable glycemic control during the hospital stay, which was true in most reviewed studies.

### 4.4. Interpretation

Mean blood glucose levels were closely associated with mortality among hospitalized COVID-19 patients with diabetes. Poor glycemic control in critically ill patients leads to an increased mortality rate, increased infection rate, mechanical ventilation, and prolonged hospitalization in critically ill patients. Patients who did so during the admission period had significantly higher glucose levels than those who were discharged (fully recovered). Our meta-analysis suggests that tighter glycemic management may positively impact COVID-19 patients, regardless of whether they have diabetes. During this coronavirus pandemic, using an aggressive glycemic control protocol could potentially improve clinical outcomes, a subject that requires further study on an extensive scale.

### 4.5. Strengths and Limitations of the Study

The inclusion of studies, regardless of the study design and where they were published or conducted, provided a comprehensive prevalence estimation of the death rate among COVID-19 patients with hyperglycemia. In addition, the applied estimates did not compromise sample size or power. A substantial sample size and relative geographic diversity (participants’ origin) may improve the generalizability of our systematic review and meta-analysis. Despite these strengths, our review and analysis had several limitations. For example, owing to the inclusion of articles published only in the English language, the obtained results might be subject to bias as other studies written in different languages were excluded. However, this case is unlikely or minimal (due to the coronavirus pandemic, the researchers and scientists aimed to reach a broader audience). Furthermore, our meta-analysis could not account for the death rates of COVID-19 patients with high BG levels who remained hospitalized after the conclusion of the study. Finally, the studies included only covered several countries where the studies were conducted; therefore, the interpretation of the findings should be performed cautiously, as one geographical region should not be generalized to the global level.

## 5. Conclusions

The main implication is that it approximates the mortality burden in admitted COVID-19 patients, particularly those with high BG. The data provided might be useful to healthcare workers and authorities when strategizing hospital management protocols for these types of patients, such as close monitoring, efficient triaging, and specialist care. Furthermore, the outlined mortality burden and health complications highlight the immediate need for additional clinical studies and research to ascertain key determinants.

## Figures and Tables

**Figure 1 jcm-12-02555-f001:**
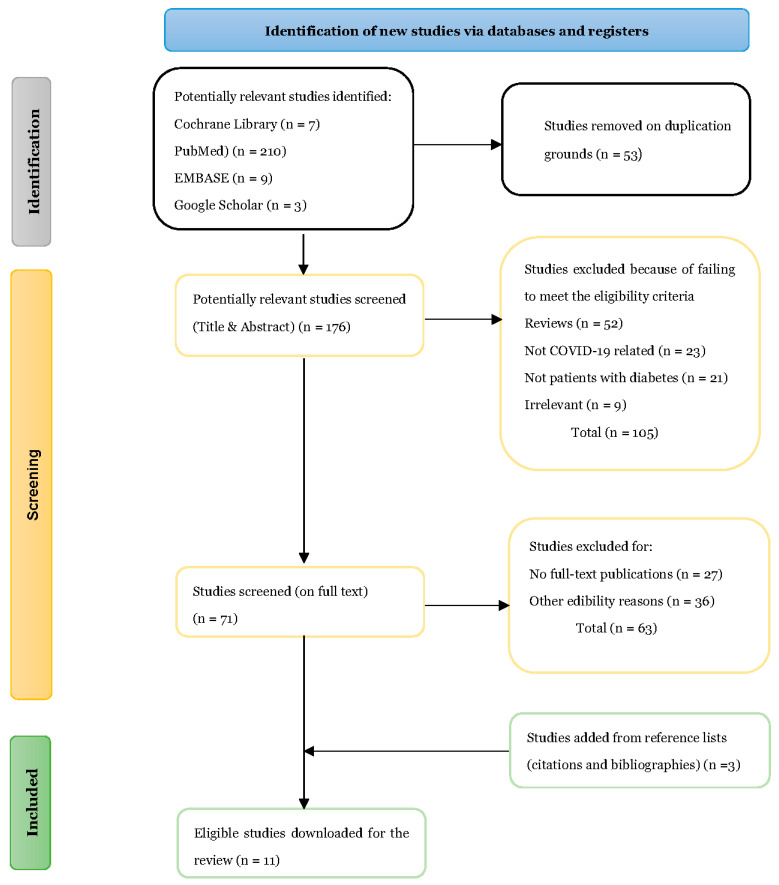
PRISMA flow diagram of the search strategy, selection process, and included studies.

**Figure 2 jcm-12-02555-f002:**
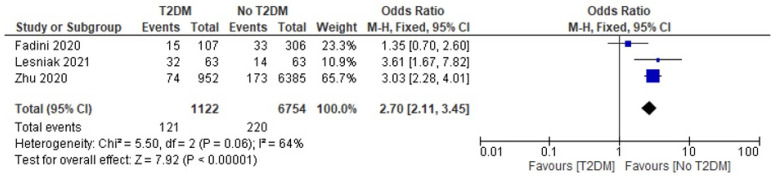
Forest plot for the outcome of death in the diabetic vs. non-diabetic COVID-19 patients’ comparison (calculated using a fixed effect OR of 2.70) [4,25,33].

**Figure 3 jcm-12-02555-f003:**
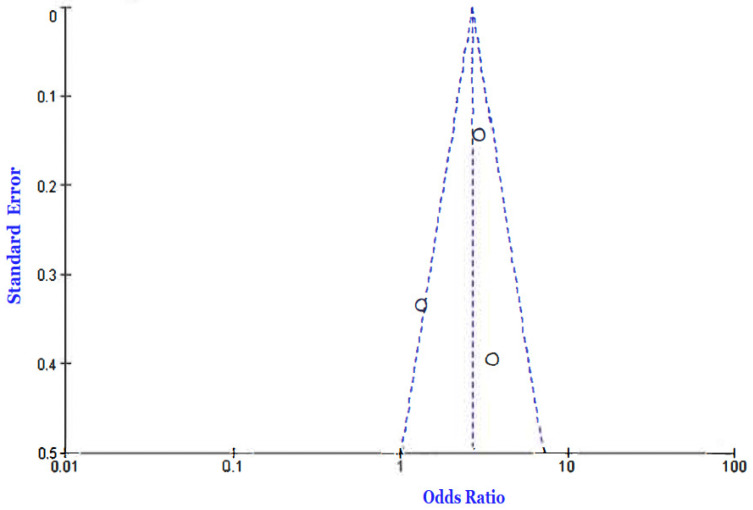
Funnel plot of the outcome of death in the diabetic vs. von-diabetic COVID-19 patients’ comparison [4,25,33].

**Figure 4 jcm-12-02555-f004:**
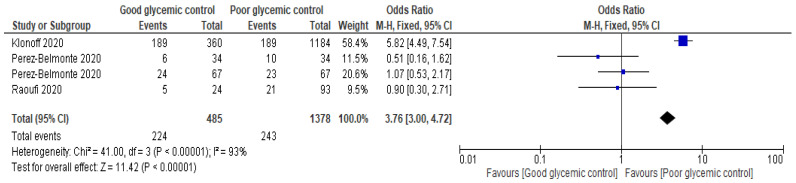
Forest plot for the outcome of death in the well-controlled vs. poorly controlled glycemic comparison (calculated using a fixed-effects OR of 3.76) [3,29,30].

**Figure 5 jcm-12-02555-f005:**
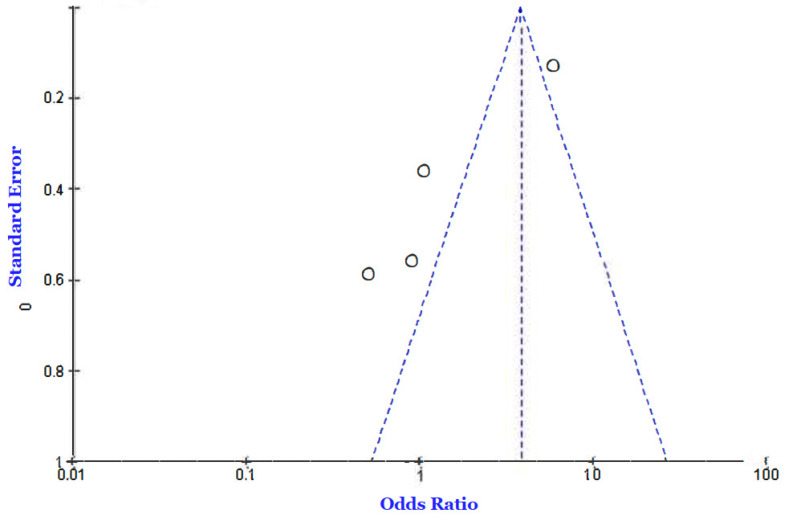
Funnel plot of the outcome of death in well-controlled vs. poorly controlled glycemic comparisons [3,29,30].

**Figure 6 jcm-12-02555-f006:**
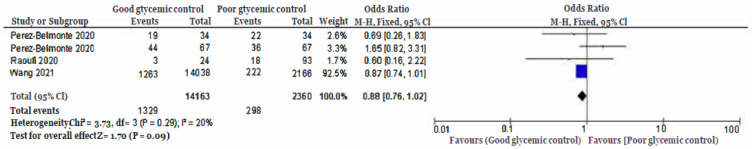
Forest plot for the outcome of complications in the well-controlled vs. poorly controlled glycemic comparison (calculated using a fixed-effect OR of 0.88) [29,30,31].

**Figure 7 jcm-12-02555-f007:**
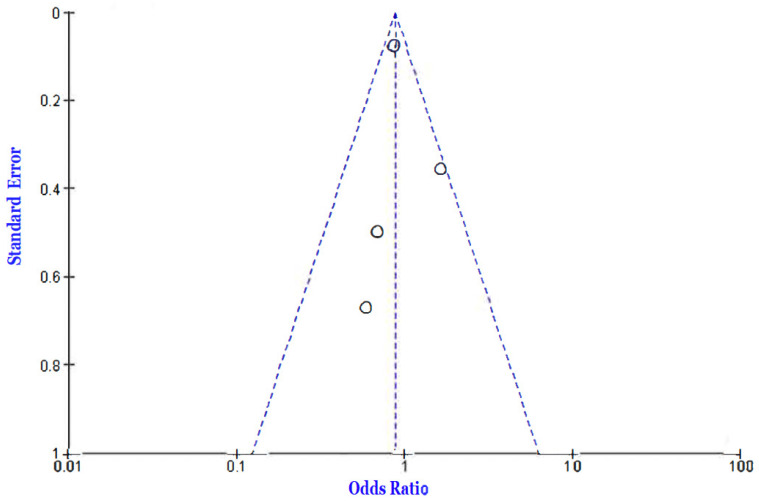
Funnel plot of the outcome of comparison in the well-controlled vs. poorly controlled glycemic comparison. [29,30,31].

**Table 1 jcm-12-02555-t001:** Summary of the basic characteristics of the included studies.

Author/Year	Study Design	Country	Participants	Intervention	Results	Conclusion
Mishra et al. (2021) [28]	Cohort	India	32	Blood glucose was monitored in CAM patients.	In patients with CAM, 87.5% had Diabetes Mellitus as the most common co-morbidity.	Uncontrolled BG in CAM patients reported high fatality.
Lesniak et al. (2021) [25]	Cohort	USA	292	Glycemic control among diabetic and non-diabetic COVID-19 patients in-hospital mortality	Significant fatality and hospitalization among COVID-19 inpatients diabetic and non-diabetic.	Study shows link between mortality and BG levels, suggesting the advantages of well-managed BG.
Raoufi et al. (2020) [29]	Cohort	Iran	117	Two groups of COVID-19 patients based on HbA1c values screened.	No difference in mortality and recovery rate between two groups (*p* = 0.54 and *p* = 0.85, respectively).	No significant difference in clinical outcomes and chest CT severity scores between the two groups.
Perez-Belmonte et al. (2020) [30]	Cohort	Spain	2666	All COVID-19 patients with type 2 diabetes mellitus on hypoglycemic medication in the Spanish Society of Internal Medicine’s registry followed for mortality	Covid patients with home glucose-lowering drugs showed a significant association with in-hospital deaths, need for ICU, mechanical ventilation and increase hospital stay.	In patients with type 2 diabetes mellitus admitted for COVID-19, at-home glucose-lowering drugs showed no significant association with mortality and adverse outcomes.
Wang et al. (2021) [31]	Cohort	USA	16,504	COVID-19 patients with DM on Insulin, metformin, ACEIs, angiotensin receptor blockers, and corticosteroids.	The HR of longitudinal HbA1c for risk of ICU = 1.12 (per 1% increase, *p* < 0.001) and 1.48 (comparing group with poor (HbA1c ≥ 9%) and adequate glycemic control (HbA1c 6–9%), *p* < 0.001).	Combination of metformin, insulin, and corticosteroids prevents COVID-19 patients with T2D developing severe complications.
Boeder et al. (2022) [17]	Cohort	USA	24	CGM in critically ill COVID-19 patient in emergency department	Individuals’ glycemic control improved when CGM was used (mean difference –30.7 mg/dL). Mean absolute relative difference of 14.8% and 99.5% of CGM.	BG management in COVID-19 patients improved during IV therapy.
Klonoff et al. (2020) [3]	Cohort	USA	1544	Patients grouped according to achieved mean glucose category in mg/dL.	In non-ICU patients, severe hyperglycemia BG >13.88 mmol on days 2–3 linked with high fatality (HR) 7.17; 95% CI 2.62–19.62) compared with patients with BG <7.77 mmol/L). In patients admitted directly to the ICU. Severe hyperglycemia on admission reported increased mortality (HR 3.14; 95% CI 1.44–6.88).	Hyperglycemia highly linked with poor outcomes in COVID-19 inpatients.
Bhatti et al. (2022) [32]	Cohort	Pakistan	638	COVID-19 patients with and undiagnosed DM.	Mortality was lowest in Patients with HbA1c of <7% (*p* < 0.001). Need for ICU admission highest in patients with HbA1c between 7 and 10% (*p* 0.002).	Effective glycemic control is related to lower risk of fatality in COVID-19 infections.
Zhang et al. (2020) [7]	Cohort	China	52	Retrospectively hospitalized COVID-19 patients with and without DM reviewed.	COVID-19 patients with diabetes were more likely to develop severe or critical disease conditions with more complications and had higher incidence rates of antibiotic therapy, non-invasive and invasive mechanical ventilation, and death (11.1% vs. 4.1%).	COVID-19 patients with diabetes showed poor clinical outcomes.
Fadini et al. (2020) [4]	Cohort	Italy	413	Retrospectively COVID patients with diabetes analyzed for composite of ICU admission or death.	ICU admission/death in 37.4% of patients with diabetes compared to 20.3% in those without diabetes. (RR 1.85; 95% CI 1.33–2.57; *p* < 0.001).	Newly diagnosed diabetes is key determinant of COVID-19 severity.
Zhu et al. (2020) [33]	Cohort	China	7337	Retrospectively studying subjects with COVID-19 and diabetes for prognosis	Subjects with DM needed more medical interventions. Higher death rates (7.8% versus 2.7%; (HR = 1.49) and multiple organ injuries than the non-diabetic patients.	DM is a key predictor for COVID-19 infection progression.

CAM: COVID-19 associated Mucormycosis, BG: Blood glucose, DM: Diabetes mellitus, HbA1C: Hemoglobin A1C, CT: Computer Tomography, ICU: Intensive care unit, RR: Relative risk, CI: Confidence interval, HR: Hazards ratio.

## Data Availability

No data required or created for this study.

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
