# Peer review of "Glycemic Control in Critically Ill COVID-19 Patients: Systematic Review and Meta-Analysis"

_jcm, 2023, doi:10.3390/jcm12072555_

Round 1

Reviewer 1 Report

1.       What are the future prospects of this study? Please discuss and include this in the abstract too.

2.       The authors should provide a proper background at the start of the abstract.

3.       Figure2. What is on the x and y axis? Please provide the complete details in the figure legend.

4.       All the figure legends should have complete details. Please include and update.

5.       Did the authors focus on the sex bias in this study? Male-female sex bias is well known in the case of COVID-19. How it is associated with the diabetes glycemic index? Is there any evidence? Please discuss.

6.       Did the authors consider the vaccination status? Please discuss. 

Reviewer 2 Report

The manuscript is interesting as well as the topic. I have some concerns to address.

First of all, the introduction should be more focused on the role of hyperglycemia in the onset of adverse outcomes beyond diabetes, please find some updated reference to cite and discuss:

- Stress Hyperglycemia Drives the Risk of Hospitalization for Chest Pain in Patients With Ischemia and Nonobstructive Coronary Arteries (INOCA). Diabetes Care. 2023 Feb 1;46(2):450-454. doi: 10.2337/dc22-0783. PMID: 36478189

- Hyperglycemia drives the transition from pre-frailty to frailty: The Monteforte study. Eur J Intern Med. 2023 Jan 10:S0953-6205(23)00011-0. doi: 10.1016/j.ejim.2023.01.006. Online ahead of print.

PMID: 36635128 

- Hyperglycemia and Physical Impairment in Frail Hypertensive Older Adults. Front Endocrinol (Lausanne). 2022 Apr 14;13:831556. doi: 10.3389/fendo.2022.831556. eCollection 2022.

PMID: 35498439

Then, Figure 1 looks unclear. I would suggest you to change the format of the text inside the boxes. An Arial/Calibri format shouuld be easier to read.

Same problem in Table 1.

There is a mistake in the number of the figure. The first forest plot should be changed in Figure 2 (instead of Figure 1) and hereafter the following figures.

Moreover, all the plot should be re-submitted in increased quality (i. e. TIFF or PNG).

Author Response

Please the attachment.

Round 2

Reviewer 1 Report

Authors successfully responded to the reviewer's comments and updated the manuscript as well.